# Predictive whisker kinematics reveal context-dependent sensorimotor strategies

**Avner Wallach**[1,2☯]*, **David Deutsch**[1,3☯], **Tess Baker Oram**[1,4☯], **Ehud Ahissar**[1]

**1** Department of Neurobiology, Weizmann Institute of Science, Rehovot, Israel, **2** Zuckerman Institute, Columbia University, New York, New York, United States of America, **3** Princeton Neuroscience Institute, Princeton University, Princeton, New-Jersey, United States of America, **4** Janelia Research Campus, Howard Hughes Medical Institute, Ashburn, Virginia, Unites States of America

☯ These authors contributed equally to this work.
* aw3057@columbia.edu

**Data Availability Statement:** Datasets and analysis files are available in the following GitHub repository: https://github.com/avner-wallach/Rat-Behavior.git.

## Abstract

Animals actively move their sensory organs in order to acquire sensory information. Some rodents, such as mice and rats, employ cyclic scanning motions of their facial whiskers to explore their proximal surrounding, a behavior known as whisking. Here, we investigated the contingency of whisking kinematics on the animal's behavioral context that arises from both internal processes (attention and expectations) and external constraints (available sensory and motor degrees of freedom). We recorded rat whisking at high temporal resolution in 2 experimental contexts—freely moving or head-fixed—and 2 spatial sensory configurations—a single row or 3 caudal whiskers on each side of the snout. We found that rapid sensorimotor twitches, called pumps, occurring during free-air whisking carry information about the rat's upcoming exploratory direction, as demonstrated by the ability of these pumps to predict consequent head and body locomotion. Specifically, pump behavior during both voluntary motionlessness and imposed head fixation exposed a backward redistribution of sensorimotor exploratory resources. Further, head-fixed rats employed a wide range of whisking profiles to compensate for the loss of head- and body-motor degrees of freedom. Finally, changing the number of intact vibrissae available to a rat resulted in an alteration of whisking strategy consistent with the rat actively reallocating its remaining resources. In sum, this work shows that rats adapt their active exploratory behavior in a homeostatic attempt to preserve sensorimotor coverage under changing environmental conditions and changing sensory capacities, including those imposed by various laboratory conditions.

## Introduction

Perception is a process in which the sensory organ is actively employed in order to acquire sensory data from the external environment [1–6]. In his classic study, Alfred L. Yarbus [1] demonstrated active sensing in human visual perception; Yarbus showed that different behavioral contexts, determined by giving subjects perceptual instructions, entail different spatial sampling strategies. A similar approach was employed to study sensorimotor exploration in other

**Funding:** EA has received funding from the European Research Council (ERC) under the EU Horizon 2020 Research and Innovation Program (grant agreement no. 786949), the Minerva Foundation funded by the Federal German Ministry for Education and Research, the United States-Israel Binational Science Foundation (BSF, grant No. 2017216), the Adelis foundation, and the Irving B. Harris Fund for New Directions in Brain Research. The funders had no role in study design, data collection and analysis, decision to publish, or preparation of the manuscript.

**Competing interests:** The authors have declared that no competing interests exist.

**Abbreviations:** CDF, Cumulative Distribution Function; GMM, Gaussian Mixture Model; LED, light-emitting diode; TIP, Touch Induced Pump.

visual animals [7,8], yet little is known about the effects of context on spatial sampling in other modalities.

Many mammals use the long hairs (vibrissae or whiskers) on either side of their snout to navigate the environment and to collect information about their proximal surroundings [9]. In some rodents, movements of the whisker array are used to actively acquire tactile information about both the position and nature of nearby objects [10]. These movements are, in turn, affected by the acquired sensory information, as well as by other "top-down" modulatory processes [11–13]. In other words, vibrissal perception is not solely active but is also reactive, giving rise to closed-loop dynamics of the perceiving organism and its environment [14–16].

Several studies have described basic components of vibrissal active behavior, both those observed in synchronous exploratory whisking in air [2,17–21] and those related to interactions with external objects [22–25]. Only a handful of studies, however, have analyzed how vibrissal behavior is affected by behavioral context. Arkley and colleagues [26], e.g., showed dramatic effects of training and environmental familiarity on the whisking strategy employed by rats. Furthermore, they showed that whisking strategy reflects the animals' expectation of future object encounters. In another study by the same group, Grant and colleagues [27] tracked the developmental emergence of previously described behaviors in rat pups' first postnatal weeks. Finally, rats change their whisking strategy in response to external perturbations, keeping some behavioral variables controlled while modulating others in order to maintain perceptual performance [28]. The effects of behavioral context on free exploratory whisking, however, remain poorly described.

In laboratory experiments, highly dominant contextual factors emerge from the experimental methodology. Experimental biologists are often forced to impose methodological constraints on their study subjects to ensure precise and stable observations that are amenable to analysis. One of the most common practices in neurophysiological and neuroimaging studies is "motion restraint," examples of which are (i) head-fixing, in which head movements are eliminated by the physical anchoring of the head [29], and (ii) body restraint, in which head movements are permitted while the body is restrained [21]. Such procedures entail drastic reduction of the motor degrees of freedom available to the animal, as well as the introduction of psychological stress [29]. An additional practice that is prevalent in the study of vibrissal perception is the reduction of the number of vibrissae available to the rodent, either by trimming or by plucking them from a full pad of 33 macrovibrissae that are arranged in 5 rows to (i) a single row [30], (ii) a few whiskers [23], or (iii) none at all [9]; in many cases, this is done to facilitate precise measurement of whisker position and shape using overhead videography. This procedure directly and selectively reduces the rodents' sensory degrees of freedom, and was shown to entail compensatory behavioral adjustments in the context of object interrogation [31]. Despite the ubiquity of such manipulations and their possible implications on the sensorimotor system, no attempt has been made so far to quantify the adaptations they might entail during exploratory behavior.

Quantification of such compensatory adaptation is essential for discriminating between open- and closed-loop models of perception. When considering an individual perceptual epoch, open-loop models assume that sensation is "presented" to the brain, which extracts the information it needs using computational tools. Closed-loop perception assumes that the generation of sensations is actively controlled as the sensory information accumulates [14,32–34]. The 2 schemes yield contradicting predictions with regard to the so-called "controlled variables" [35]: closed-loop perception predicts that there will be motor-sensory variables that are maintained invariant despite external or embodied constrains, while open-loop perception predicts that such variables should not exist [14,35]. Our experiments and analyses are aimed at directly testing these predictions.

Here, we compare different aspects of vibrissal behavior measured in 3 contexts (Fig 1A): (i) head-fixed rats with a trimmed whisker pad (in this case, only 3 caudal macrovibrissae—C1, C2, and D1—were untrimmed on either side; reuse of data published in [23]), (ii) freely moving rats with the same 3-whisker configuration as in (i), and (iii) freely moving rats with an entire single row (row C) of macrovibrissae on either side (termed here "free single-row," for the sake of brevity; behavioral apparatus is illustrated in S1 Fig). We focused only on segments in which the animals performed exploratory rhythmic movements in free air without encountering any object with the whiskers. It should be noted that whisker contacts with the floor could not be ruled out in the freely moving rats because of video limitations (resolution and focus). However, based on a previous study [13], we can estimate the probability of such floor contacts to be only 2.5% per cycle for any whisker, and therefore, they are not expected to significantly alter our findings. Moreover, head-fixed rats were positioned such that no floor contacts were possible. We begin by describing the different parameters of whisking behavior used in our analyses, and then focus our analysis on a kinematic feature called the "free-air pump" [20,36]. We then show that head-fixing exerts a dramatic shift in the whisking profile, suggesting an adaptive redistribution of sensorimotor exploratory resources. Finally, we demonstrate that whisker trimming likewise entails an alteration in whisking strategy.

## Results

### Characterizing whisking behavior

All analyses were performed on free-air exploratory whisking, in the absence of interactions with external objects. It has been shown that this mode of whisking is characterized by extremely high correlations between the angle of motion of the different whiskers [19,23], i.e., the entire whisker pad moves in unison. Since all whiskers move together, analyses of free-air whisking can be performed on one representative whisker; whisker C2 is often chosen because it is centrally located within the whisker pad [37]. In the analysis of head motion, we used whisker C2 on both sides of the face; in all other analyses, only the left side was used. Our tracking tool extracted from the videos the whisker's "base angle," i.e., the angle of the whisker at the point it enters the skin relative to the line connecting the ipsilateral eye and the tip of the nose. To quantify the governing variables of exploratory whisking, we employed "rhythmic decomposition" on each of the tracked segments [21]; this algorithm models whisking dynamics as rapid oscillations (whisking cycles), modulated by a multiplicative "amplitude process" and an additive "offset" process (see Fig 1B and Methods). The whisking amplitude was bimodally distributed (Fig 1C and 1D), consisting of either high-amplitude "bouts" (non-shaded in Fig 1B, light-colored lines in Fig 1C and 1D) or low-amplitude "pauses" (shaded and hatched in Fig 1B, dark-colored lines in Fig 1C and 1D); it has been suggested that these 2 modes reflect 2 attentional states: an active exploratory state during bouts and a passive, "receptive" state during pauses [16,38]. By fitting a Gaussian Mixture Model (GMM), we found the maximum-likelihood threshold between these modes (2.6 deg for head-fixed, 2.1 deg for freely moving; see arrows in Fig 1C and 1D); we therefore set a conservative threshold of 2.5 deg for all data sets. Overall, head-fixed rats whisked less frequently than freely moving, with head-fixed rats pausing for 27% of the tracked time, while freely moving rats paused 7.8% of the tracked time (inset, Fig 1B). This reflects the well-known reluctance of head-restrained rats to whisk; often, some sensory stimulation (e.g., olfactory) is required to encourage head-fixed animals to whisk. The duration of whisking bouts varied greatly (Fig 1E, Cumulative Distribution Function [CDF]); in the head-fixed data set, the median bout duration was 0.366 s (2 whisking cycles), while the mean was 0.92 s (5.86 cycles) ($N = 363$). This large difference between median and mean reflects the "heavy tail" of the bout distribution. Indeed, the bout

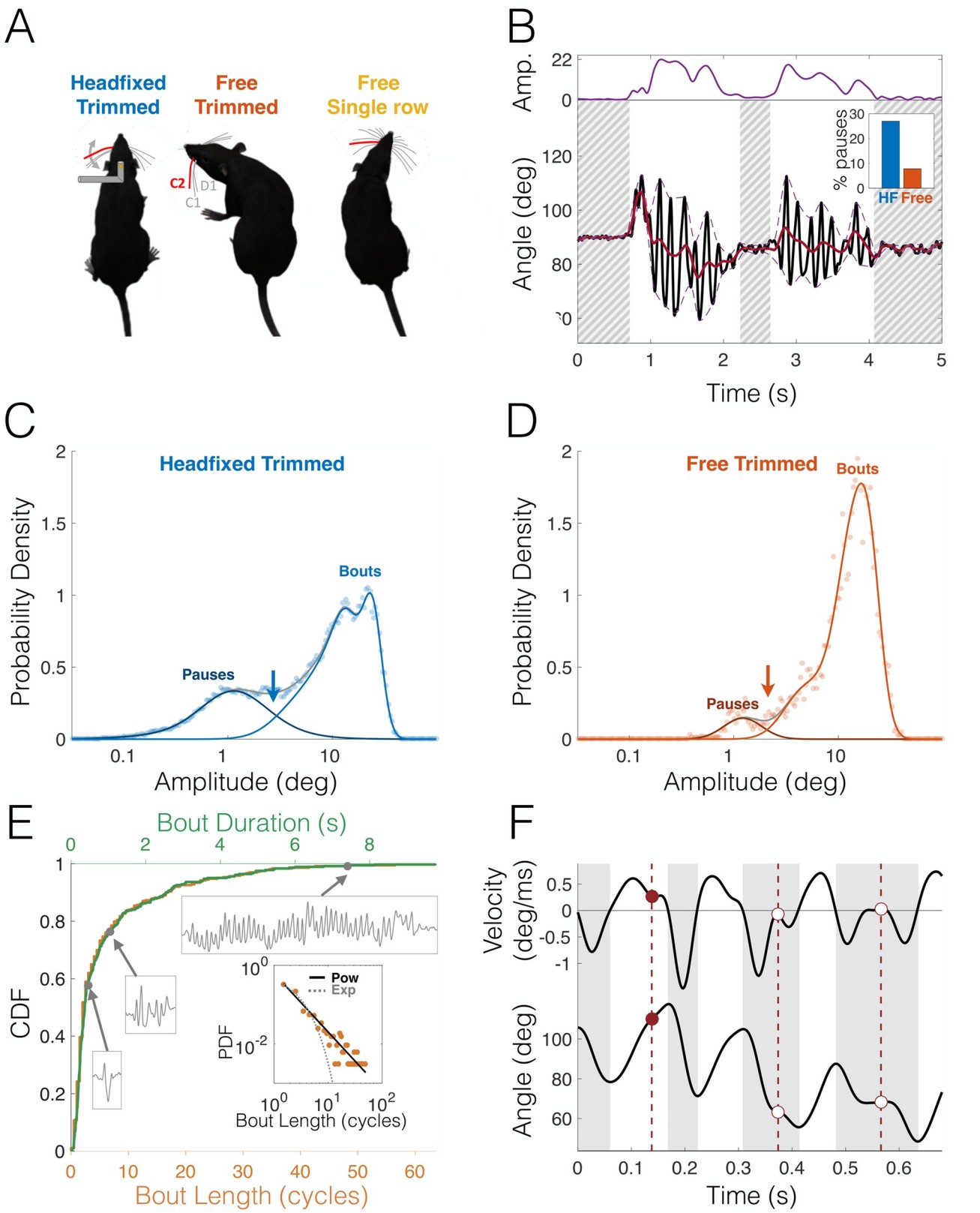

**Fig 1. Methodology, whisking bouts, and free-air pumps.** (A) Three data sets were used in this study: HF rats with only whiskers D1, C1, and C2 (left); freely moving rats with only whiskers D1, C1, and C2 (free trimmed, center); and freely moving rats with 7 whiskers of row C (C1–C7) (free single-row, right); only motion of C2 (in red) was used in all analyses; only the left side whisker was used in all analyses except in Figs 3 and 4. (B) Rhythmic decomposition of whisking. Dynamics of the whisker's base angle (black line) were analyzed as fast oscillations modulated by multiplicative amplitude (top, purple) and additive offset (black). Hatched: nonwhisking epochs (amplitude < 2.5 deg). Inset: HF animals spend less time whisking. (C,D) Whisking amplitude histogram (circles; note the logarithmic scale) for the HF rats (C) and freely moving (D) trimmed rats. Histograms were fitted using a GMM (gray line). Distributions exhibited 2 modes: low-amplitude pauses (dark-colored lines) and high-amplitude bouts (light-colored lines). Arrows: maximum-likelihood thresholds (2.6 deg for HF, 2.1 deg for freely moving). (E) CDF of bout duration in cycles (bottom abscissa, orange) and in seconds (top abscissa, green) in HF rats. Examples of bouts 0.44 s, 1.04 s, and 7.3 s long are shown. Inset: probability density histogram of bout length is heavy-tailed; solid black line: power-law fit, dotted line: best exponential fit, shown for comparison. (F) Example of a whisking trajectory. Bottom: base angle; top: angular velocity. White background: protraction phases; gray background: retraction phases. Red markers and vertical dashed lines: free-air pumps, phases in which velocity profile is double-peaked; filled circle: protraction pump, empty circles: retraction pumps. The data and analysis code for this figure can be found here: https://github.com/avner-wallach/Rat-Behavior.git. CDF, Cumulative Distribution Function; GMM, Gaussian Mixture Model; HF, head-fixed; PDF, Probability Density Function.

probability density is well fitted with a power-law function (inset of Fig 1E). The freely moving data sets contained much shorter segments since object contacts had to be edited out (head-fixed segments were 7 ± 3.1 s long, while freely moving segments were 2.6 ± 1.7 s long). This, together with the scarcity of pauses, led to there not being enough freely moving complete bouts (in which both the beginning and the end of the bout are recorded) to allow for a statistical analysis (42 freely moving bouts versus 363 head-fixed bouts). Most of the analyses described in this paper focused on features related to the active bout epochs only; e.g., the whisking pumps (see below) are only defined in the context of active protractions or retractions.

Each whisking cycle is composed of 2 stages: protraction, in which the whiskers move rostrally (i.e., when the whiskers' angular velocity is positive; unshaded in Fig 1F), and retraction, in which they move caudally (i.e., when angular velocity is negative; shaded in Fig 1F); henceforth, we will refer to protractions and retractions as the 2 "phases" of the whisking cycle. Usually, both protractions and retractions display smooth, ballistic-like kinematic profile with a single velocity peak; however, it was noted in several previous studies that some protraction/retraction phases exhibit a multipeaked velocity profile, indicating that the whisker motion consists of 2 or more consecutive "thrusts." This feature, termed a whisking "pump" (red markers in Fig 1F [20,36]), was suggested to serve as a fast "error-correction" mechanism for the whisking kinematics [20]; another hypothesis is that these pumps cause the whiskers to linger and "resample" a spatial locus of interest, thus increasing the sensory information throughput from such loci. Consistent with this hypothesis, such pumps often occur immediately (latency < 18 ms) after the whisker contacts an external object. These "Touch Induced Pumps" or TIPs [23] were shown to be related to object-oriented spatial attention [13]. We detected the occurrence of pumps in our "free-air" data, in which no objects were contacted, by first segmenting the whisking signal into phases and then finding the phases in which the velocity profile had more than one peak (see Methods). In the next section, we compare various properties of these "free-air pumps" and TIPs.

## Free-air pumps are temporally clustered and entail phase prolongation and offset shifting

TIPs cluster in time [23]. We tested the tendency of free-air pumps to cluster by measuring the cross-correlation of pump instances between cycles. The probability of free-air pump occurrence was significantly correlated across multiple cycles (Fig 2A$_{1,2}$). Importantly, these correlations were specific to pumps of the same whisking phase: a cycle in which a pump occurred in the protraction phase was likely to be preceded and followed by other cycles with protraction pumps ($p < 0.05$ up to lags of 7 cycles, random permutations; Fig 2A$_1$), and the same was observed for retraction pumps ($p < 0.05$ for up to and beyond 10 cycles, random

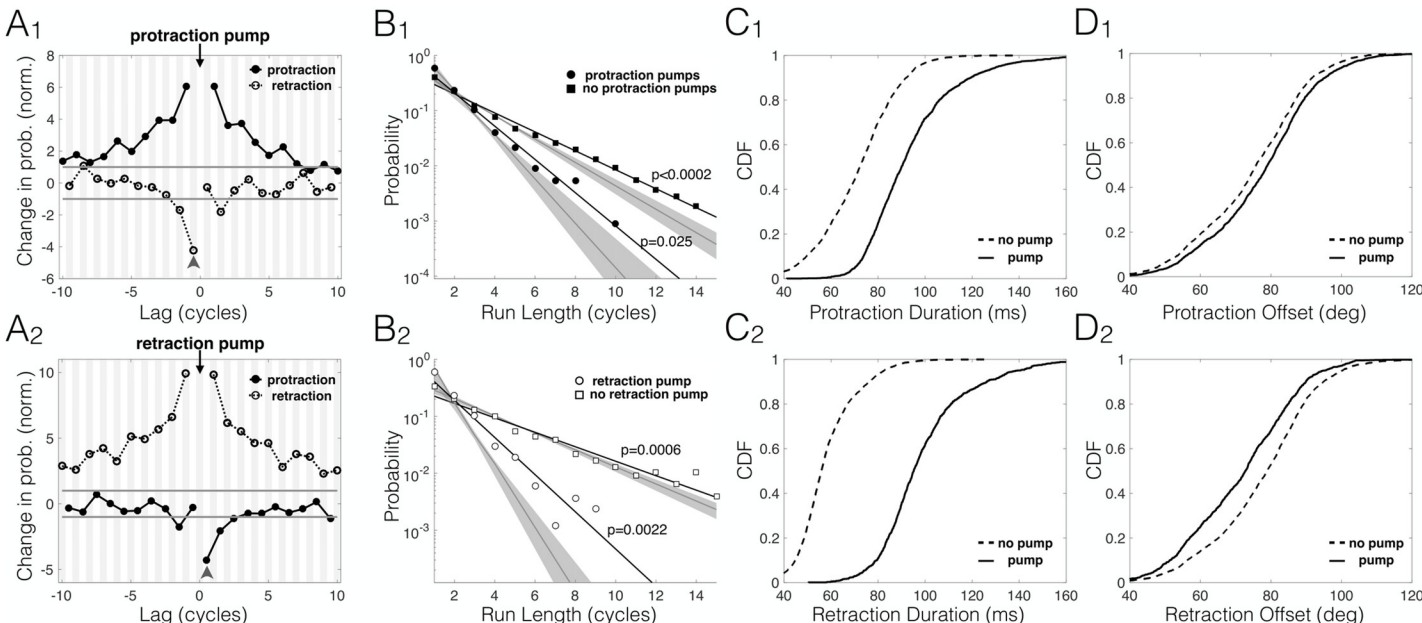

**Fig 2. Phases with free-air pumps are clustered in time.** ($A_{1,2}$) Temporal clustering. Change in pump probability (normalized to the 0.05 significance level; i.e., −1 is the fifth percentile and +1 is the 95th percentile, both marked with a gray horizontal line) in protractions (solid, filled circles, white background) and retractions (dashed, empty circles, gray background), given that a pump (black arrow) occurred in protraction ($A_1$, top) or retraction ($A_2$, bottom) of a cycle, as function of time lag from that cycle. Same-phase pumps are strongly correlated across multiple cycles. Different-phase pumps are overall uncorrelated, though a retraction pump is unlikely to be immediately followed by a protraction pump (gray arrowheads). ($B_{1,2}$) Probability distributions of series of consecutive pumps (circles) and no pumps (squares) of different run length, shown separately for protractions (filled markers, $B_1$, top) and retractions (empty markers, $B_2$, bottom). Long sequences of pumps/no pumps are significantly more common than chance level because of the temporal correlations shown in A. Gray shading: 0.95 confidence margins for controls generated by random permutations. ($C_{1,2}$) Phase prolongation. CDF of phase duration for protractions ($C_1$, top) and retractions ($C_2$, bottom) with pumps (solid) and without pumps (dashed). Phases with pumps are longer (means: protractions 93 ms/71 ms, retractions 99 ms/58 ms with/without pumps, respectively). ($D_{1,2}$) Offset shifting. CDF of mean whisking offset of protractions ($D_1$, top) and retractions ($D_2$, bottom) with pumps (solid) and without pumps (dashed). The offset during phases with pumps are shifted towards the direction of motion, i.e., forward in protractions and backwards in retractions (means: protractions 77.7 deg/74.6 deg, retractions 77.3 deg/71.5 deg with/without pumps, respectively). The data and analysis code for this figure can be found here: https://github.com/avner-wallach/Rat-Behavior.git. CDF, Cumulative Distribution Function.

permutations; Fig 2$A_2$). Sequences of consecutive protraction/retraction pumps were significantly more frequent than those obtained by random permutations ($p < 0.0002$ and $p = 0.0006$; circles, Fig 2$B_{1,2}$), and the same was true for sequences of no pumps ($p = 0.025$ and $p = 0.0022$; squares, Fig 2$B_{1,2}$). Retraction pumps were less likely to be immediately followed by protraction pumps ($p < 0.005$, random permutations; gray arrowheads in Fig 2$A_{1,2}$), but otherwise, pumps of opposing phases were uncorrelated.

Did cycles containing free-air pumps differ from those lacking them? In the context of object encounters, protractions containing TIPs are longer and shifted forward when compared with protractions with no pumps [13,23]. This was also true for free-air pumps: first, phases with such pumps were much longer than those lacking pumps (31% longer in protraction and 71% longer in retraction; Fig 2$C_{1,2}$; as previously shown in [20]). Second, the whisking offset (i.e., the midpoint angle of the motion) was shifted in the direction of the whisking phase that included the pumps, i.e., more protracted in protractions that included pumps and more retracted in retractions that included pumps (+4.2% and −7.6% change in protraction/retraction, respectively; Fig 2$D_{1,2}$). We note, however, that the statistical significance of this last finding is difficult to assess because of temporal correlations in the offset and pump-rate signals, which render neighboring samples statistically dependent.

To conclude, we have shown some basic properties of free-air pumps: First, free-air pumps of each whisking phase (protraction/retraction) exhibited strong temporal correlations and

therefore occurred in temporal "clusters," rather than appearing randomly and uniformly in time. Second, phases containing free-air pumps were distinct from those that lacked pumps in both their duration and offset. Importantly, these properties are similar to those previously described for TIPs [23], which were later shown to be related to object-oriented spatial attention [13].

## Direction-specific correlations between free-air pumps and head motion

We decomposed head motion into 3 components: "turn" (rotation around the midpoint between the eyes), "thrust" (longitudinal translation or forward/backward motion), and "slip" (transverse translation or side motion). While turning was uncorrelated with thrust (Pearson coefficient R = −0.035), it was highly correlated with slip (Pearson coefficient R = 0.863, see S2 Fig); these correlations reflect the fact that the axis of head rotation (the neck) is caudal to the eyes (which were the feature tracked in our analysis). Therefore, we limit our analysis of head motion to the turn and thrust variables. The likelihood of protraction pumps was correlated with contralateral head turns: turns to the left increased the pump frequency of the right whisker pad while decreasing that of the left whisker pad and vice versa (Fig 3A; plots are normalized to overall pump rates; statistical significance measured using random permutations, $N = 5,000$). We note that when the head is turning in a certain direction, the contralateral side of the snout moves forward and the ipsilateral side moves backwards, and therefore, Fig 3A demonstrates that protraction pumps on either side of the face occur more frequently when that side moves forward. Correspondingly, protraction pumps were also more frequent during forward thrust at moderate speeds (i.e., walking, but not running) when both sides of the face move forward (Fig 3B). In contrast, retraction pumps occurred mostly when the rat kept its head motionless (close to zero velocity of turn and thrust) and were significantly inhibited during rapid motion in any direction. Therefore, head motion was accompanied with direction-specific modulations in the occurrence of free-air pumps; protraction pumps occurred mostly during forward motion (either of the entire head or of the side containing the pumping whisker), whereas retraction pumps were frequent when the head stayed nearly motionless.

## Free-air pumps are predictive of changes in head motion

We next checked whether free-air pumps systematically precede changes in the animal's head motion. We performed event-triggered analysis to explore the reciprocal relations between head and pump dynamics. First, we identified motion-change events, in which the rat changed the direction of head turning or thrust. We then measured the dynamics of free-air pump rate, relative to the onset of the change (the onset of acceleration in the opposite direction). Onset of forward motion was preceded by a substantial increase in protraction pumps, peaking around 100 ms prior to the change ($p < 2 \times 10^{-4}$, random permutations; black arrow, Fig 4A). No such change occurred in the onset of backward motion ($p = 0.22$, random permutations; Fig 4B). Note that in both cases, retraction pumps were inhibited before and after the change occurred. Similarly, ipsilateral head turns (towards the pumping-whisker side) were preceded by an increase in protraction pump rate ($p < 2 \times 10^{-4}$, random permutations; black arrow, Fig 4C), while no such increase is seen prior to contralateral turns ($p = 0.26$, random permutations; Fig 4D).

The predictive power of free-air pumps regarding future spatial targets of the rat is confirmed by measuring the dynamics of head turning probability relative to the time a pump occurred. Consistent with the previous analysis, the average protraction pump was followed by an increased probability of turning towards the pumping-whisker side ($p = 0.0038$, random permutations, left ordinate in Fig 4E), which is also evident in the dynamics of angular

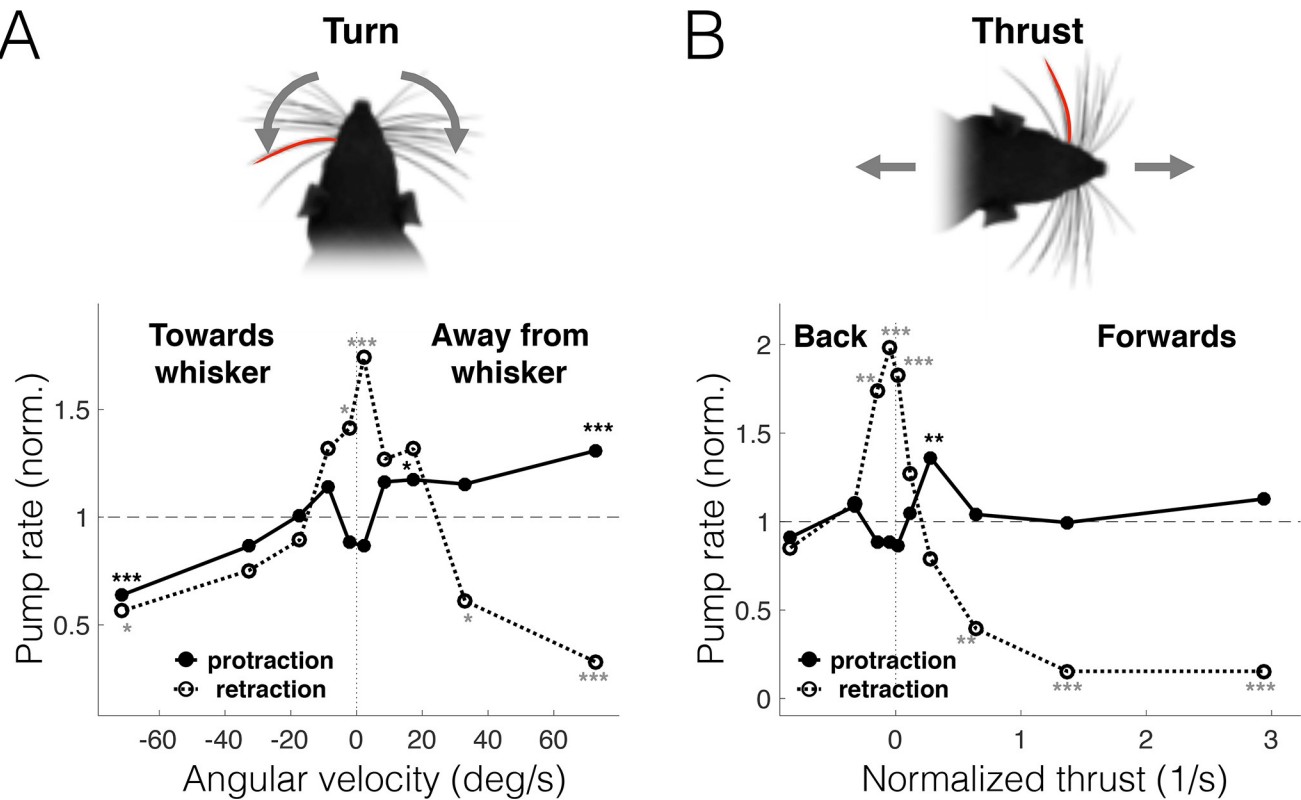

**Fig 3. Free-air pumps are related to head motion.** Pump rate (normalized to mean pump rate across all cycles) conditioned on head-motion variables in freely moving rats. Motion of whisker C2 from both sides of the snout was used in these analyses. Protraction: solid line with filled circles, retraction: dashed line with empty circles. (A) Turns. Protraction pumps are more frequent in contralateral turns, i.e., when the pumping-whisker side moves forward. Retraction pumps are more frequent when the rat does not turn its head. (B) Thrust. Protraction pumps are more frequent in low-velocity motion forward, while protraction pumps are more frequent when the rat is close to stationary. All linear velocities are normalized to head size. $^{*}p < 0.05$; $^{**}p < 0.01$; $^{***}p < 0.005$; no symbol, not significant. The data and analysis code for this figure can be found here: https://github.com/avner-wallach/Rat-Behavior.git.

acceleration ($p > 0.05$, random permutations, right ordinate in Fig 4E). Retraction pumps, however, were followed by a significant drop in this probability ($p = 0.02$, random permutations, Fig 4F). These results are not sensitive to the choice of motion-change detection threshold (see S3 Fig). We conclude that free-air pumps predict the rats' motion targets: initiation of a forward motion or a head turn were preceded by an increase in bilateral or ipsilateral protraction pumps, respectively, and a drop in retraction pumps.

## Head-fixed effects on whisking resemble those of voluntary motionlessness

Are voluntary and imposed motionlessness similar in their effects on whisking kinematics? To answer this question, we compared the frequency of protraction and retraction pumps in head-fixed rats with those observed in freely moving rats at different motion speed ranges (Fig 5A). When the entire data set of freely moving episodes is analyzed (maximal head velocity = "∞"), head-fixed rats exhibited 22% fewer protraction pumps than freely moving rats ($p = 10^{-4}$, bootstrap, $N = 4{,}056$ and 412 cycles for head-fixed and free trimmed, respectively). In contrast, retraction pumps were dramatically more prevalent in head-fixed rats (208%, $p < 10^{-4}$, bootstrap, $N = 4{,}046$ and 549 cycles for head-fixed and free trimmed, respectively). However, as we limit our analysis of pump frequency in freely moving rats to phases in which head velocity did not exceed a certain bound, the frequency of protraction pumps decreases

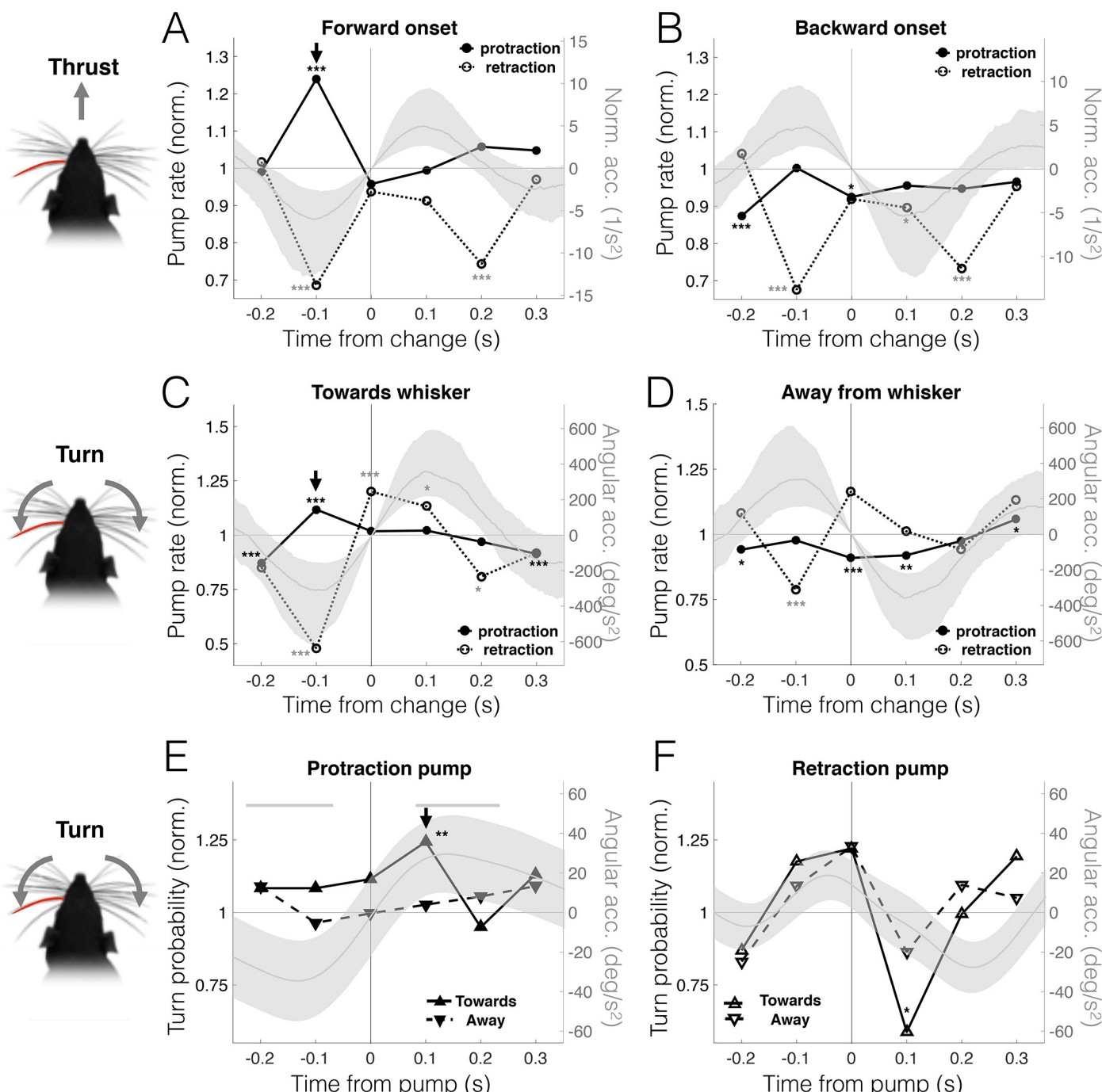

**Fig 4. Free-air pumps predict head motion.** Dynamics of pump rate (normalized to mean rate) aligned to onset of head motion (i.e., zero-crossing of acceleration). All probabilities were calculated in 100-ms wide bins. (A–B) Protraction pump rate (left ordinate) is significantly elevated 100 ms prior to onset of forward motion ($p < 2 \times 10^{-4}$, random permutations). No such increase is present in onset of backward motion ($p = 0.22$, random permutations). Right ordinate: longitudinal acceleration (normalized to head size); median: dark gray curves, interquartile range: shaded gray area. (C–D) Protraction pump rate (left ordinate) is significantly elevated 100 ms prior to head turn toward pumping-whisker side ($p < 2 \times 10^{-4}$, random permutations). No similar increase is present in onset of turn away from pumping-whisker side ($p = 0.26$, random permutations). Right ordinate: angular acceleration; median: dark gray curves, interquartile range: shaded gray area. (E–F) Ipsilateral turn probability (left ordinate) is elevated 100 ms after the occurrence of a protraction pump ($p = 0.007$, random permutations) but depressed 100 ms after a retraction pump ($p = 0.02$, random permutations). Right ordinate: angular acceleration; median: dark gray curves, interquartile range: shaded gray area. Thick gray line: statistically significant times in which average acceleration was nonzero ($p < 0.05$, random permutations). $^{*}p < 0.05$; $^{**}p < 0.01$; $^{***}p < 0.005$; no symbol, not significant. The data and analysis code for this figure can be found here: https://github.com/avner-wallach/Rat-Behavior.git.

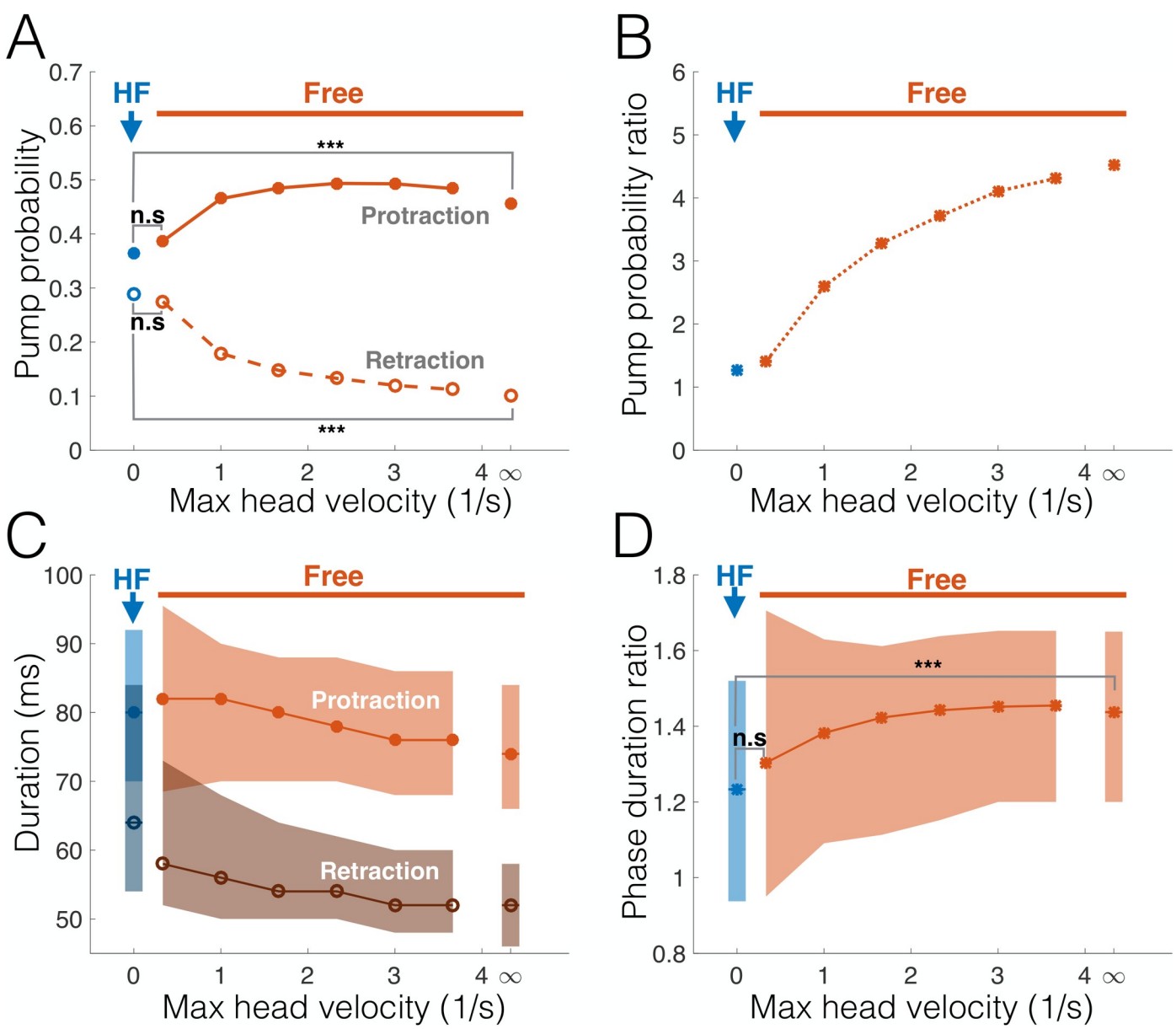

**Fig 5. Free-air pumps in HF rats resemble those in voluntary motionlessness.** Abscissa in all panels: maximal head velocity for samples analyzed from the freely moving data set (i.e., only cycles in which head velocity did not exceed this value were taken); ∞ = entire data set. (A) Probability of protraction (filled circles) and retraction (empty circles) pumps for HF (blue) and freely moving (red) rats. (B) Ratio between protraction and retraction pump probabilities. (C) Distributions of protraction (median: filled circles; interquartile range: light shading) and retraction (median: empty circles; interquartile range: dark shading) durations for HF (blue) and freely moving (red) rats. (D) Protraction-retraction duration ratio for HF (blue) and freely moving (red) rats (median: asterisks; interquartile range: shading). $^{***}p < 0.005$. The data and analysis code for this figure can be found here: https://github.com/avner-wallach/Rat-Behavior.git. HF, head-fixed; n.s., not significant.

and that of retraction pumps increases to the point of becoming statistically similar to one another ($p = 0.14$, bootstrap, $N = 75$ protractions and 80 retractions), as well as to the frequencies measured in head-fixed rats ($p = 0.21$ and 0.81 for protractions and retractions, respectively, bootstrap; Fig 5A). In other words, the overall ratio of protraction/retraction pumps drops as velocity decreases, approaching that of head-fixed rats at near motionlessness (Fig 5B). A similar trend can be seen in the distribution of protraction/retraction durations (Fig

5C). Protractions were significantly longer than retractions for the entire freely moving data set (means 74 ms and 52 ms, respectively), consistent with previous observations [17,20,39–43]. However, while both protractions and retractions increased in duration as the maximal speed decreased, the ratio of protraction duration to retraction duration decreased, approaching that of head-fixed rats at near motionlessness (Fig 5D). We conclude that freely moving rats, which display a strong emphasis on the protraction phase during motion, approach the near-parity in protraction/retraction that is typical of head-fixed rats as their velocity decreases towards motionlessness.

## Whisking envelope spatial distribution is dispersed because of head-fixing

Our findings so far suggest that limiting the rat's degrees of motor freedom (by head-fixing) entails a more even distribution of the animal's sensorimotor resources between protraction and retraction; this was reflected in features such as pump occurrence and phase duration. Can we see a similar effect in the "whisking envelope"—the combination of amplitude and offset (see Fig 1B) that dictates the range of angles covered by whisking at each cycle? It is important to note that while amplitude and offset may be, in principle, uncorrelated, they are necessarily statistically dependent because the maximal possible amplitude is always dictated by the distance from the current offset to the maximal whisker protraction and retraction angles. The theoretical domain of all possible amplitude-offset combinations, therefore, is bound by an isosceles triangle (Fig 6).

Utilization of this domain was indeed affected by head-fixing. Comparing rats with the same whisker arrays (head-fixed and freely moving trimmed rats) revealed that while freely moving trimmed rats had a restricted focal region in which they preferentially whisked (i.e., the whisks were narrowly distributed around a preferred amplitude of 11.8 deg and a preferred offset of 74.5 deg; Fig 6A), head-fixed rats exhibited a highly dispersed distribution that covered a large portion of the triangular domain (Fig 6B). This dispersion may reflect compensation for the lost motor degrees of freedom of the head and body. Indeed, whisking envelope distributions became dispersed when changes in head angle were considered in freely moving rats (relative to the mean head angle in each tracked segment, Fig 6C). To quantify the envelope dispersion in each scenario, we measured the distributions' information entropy; to evaluate statistical significance, the bootstrap method was used to generate random subsets of equal size from each data set (Fig 6D; see Methods). The entropy was significantly smaller in freely moving trimmed rats than in head-fixed ones when only whisker angle was used ($p < 10^{-4}$, bootstrap), reflecting the increased dispersion of the latter. However, no significant difference was measured when the head's degree of freedom was taken into account ($p = 0.13$, bootstrap). We conclude that head-fixed rats' whisking patterns were much more diverse than those of freely moving rats. This suggests that the rats compensated for the loss of motor degrees of freedom due to head-fixing by employing a wider range of whisking configurations.

## Whisker trimming entail shifts in whisking strategy

We next examined the impact of whisker trimming on the distributions of individual whisking variables (offset and amplitude, diluted to avoid temporal correlations; see Methods). As described above (see Fig 1A), freely moving rats had either a configuration of 3 caudal whiskers or a full row of 7 whiskers (i.e., including both caudal and rostral whiskers). Comparison of the whisking amplitude and offset distributions of these 2 data sets of freely moving rats revealed that the preferred whisking strategy was noticeably different. The trimming-induced effect consisted of 2 adjustments: first, a significant 33.8% decrease in the variance of offsets ($p < 10^{-4}$, bootstrap, $N = 292$ and 280 for single-row and trimmed, respectively; Fig 7A) while

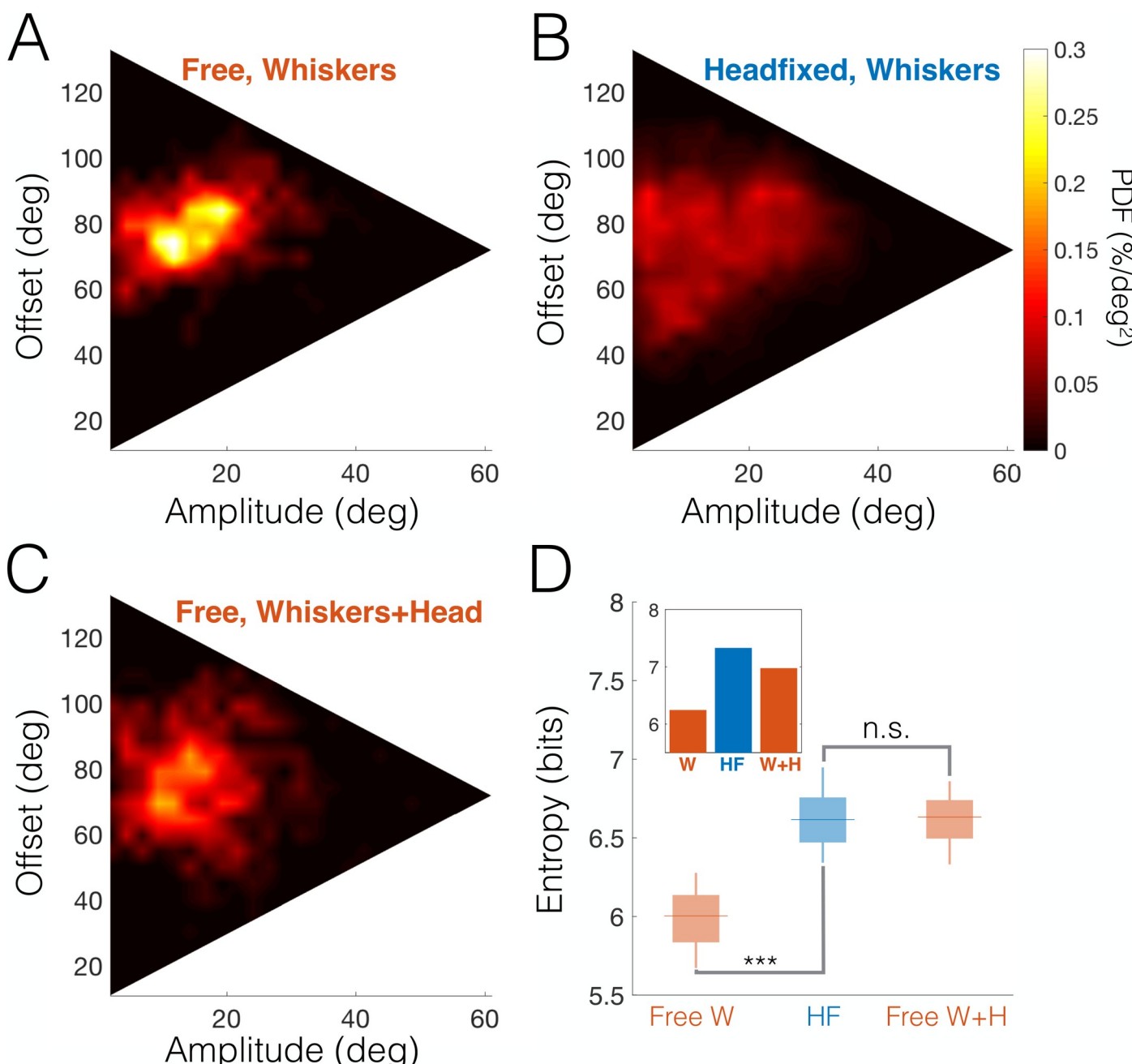

**Fig 6. HF whisking envelope distribution compensates for lost degrees of freedom.** (A–B) Bivariate probability density distributions of whisking amplitude and offset (relative to head) for freely moving (A) and HF rats (B). While freely moving rats exhibit a preferred subspace of amplitude-offset combinations, HF rats cover much of the available envelope space (range of offsets is determined by the absolute bounds on whisker C2 angle in all data sets, 11–133 deg; maximal possible amplitude is offset dependent and peaks at the median offset at 61 deg). (C) Probability distribution of whisking envelope, taking into account head rotations, for free rats. (D) Whisking envelope information entropies for random subsets taken from each data set (box-and-whisker plots: horizontal line, median; box, IQR). Mean entropy is significantly smaller for freely moving rats (Free W, left) than for HF rats (center), indicating that the whisking envelope distribution is much more dispersed during head-fixing; however, there is no significant difference when head rotations are included in the analysis (Free W + H, right). $***p < 0.005$. Inset: entropies calculated for entire data sets (free whiskers only 6.24 bits, HF 7.325 bits, free whiskers + head 6.97 bits). The data and analysis code for this figure can be found here: https://github.com/avner-wallach/Rat-Behavior.git. HF, head-fixed; IQR, Interquartile Range; n.s., not significant; PDF, Probability Density Function.

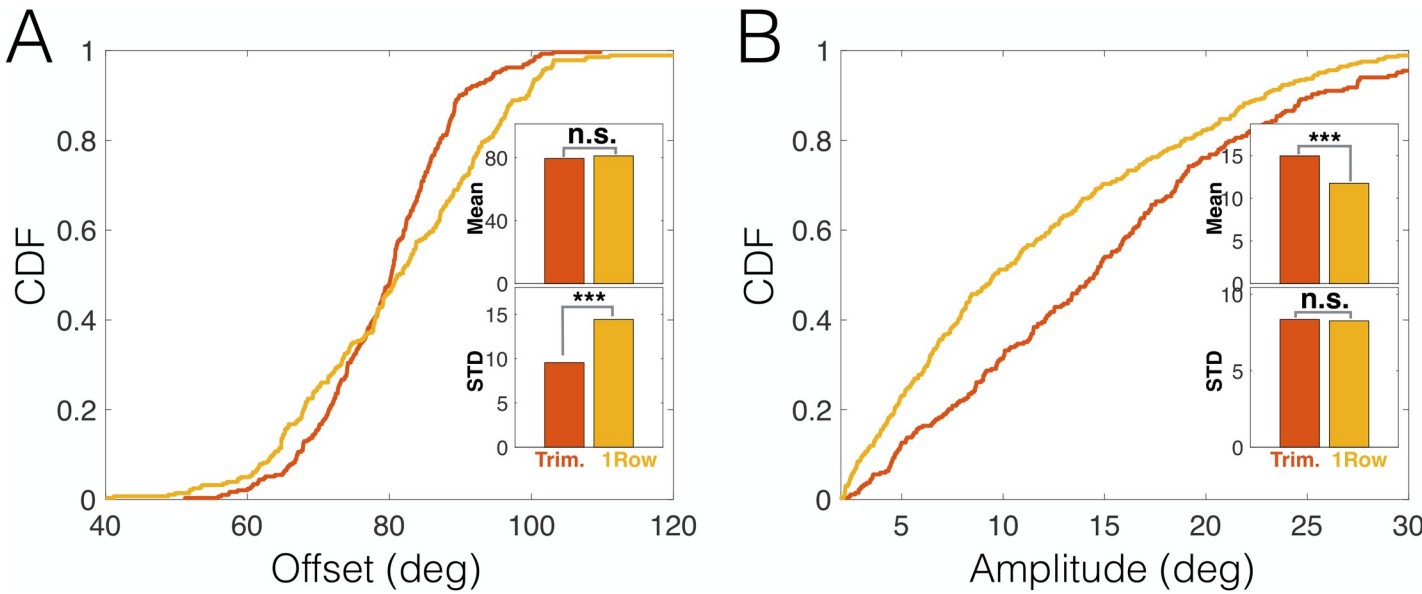

**Fig 7. Contextual shifts in whisking strategy.** (A) CDFs of whisking offset of freely moving trimmed (red) and single-row (yellow) rats. Insets: mean and STD of distributions. (C) CDFs of whisking amplitude of freely moving trimmed (red) and single-row (yellow) rats. Insets: mean and STD of distributions. ***$p < 0.005$. The data and analysis code for this figure can be found here: https://github.com/avner-wallach/Rat-Behavior.git. CDF, Cumulative Distribution Function; n.s., not significant; STD, Standard Deviation.

maintaining the mean offset almost unchanged (2% decrease, $p = 0.11$, bootstrap); and second, a significant 27.4% increase in mean amplitude ($p = 10^{-4}$, bootstrap, $N = 367$ and 268 for single-row and trimmed, respectively; Fig 7B) while not significantly changing its variance ($p = 0.91$, bootstrap). Therefore, the trimmed rats tended to employ large amplitude whisks around a relatively constant offset angle, while those having a full row of whiskers used smaller amplitudes while shifting the offset over time.

## Head-fixing entails a backward shift in offset during bouts, but not during pauses

Lastly, we compared the offset distributions in head-fixed and freely moving rats during both active whisking bouts and passive, low-amplitude pauses (see Fig 1C and 1D). When only bouts were analyzed, head-fixing caused an increase in offset variance (23.5% increase, $p < 10^{-4}$, bootstrap, $N = 1,010$ and 494 for head-fixed and free, respectively; Fig 8A), consistent with our findings above (see Fig 6). Additionally, head-fixed rats whisked around a slightly retracted offset angle (6.3% reduction, $p < 10^{-4}$, bootstrap). This further suggests that head-fixed animals explored more retracted positions than free animals, in line with the pump results described above. During pauses, however, the head-fixed and free offset distributions were not significantly different in either mean or variance ($p = 0.4$ and $p = 0.96$, respectively; bootstrap, $N = 556$ and 125 for head-fixed and free, respectively; Fig 8B). A likely explanation for this is revealed by comparing the distributions of head-motion variables (Fig 8C: absolute translation velocity; Fig 8D: absolute angular velocity) during bouts and pauses; both variables show that when rats stopped whisking, their head was nearly motionless ($p < 10^{-4}$, bootstrap, $N = 125$ and 494 for pauses and bouts, respectively). Therefore, the similarity in pause offset distribution between head-fixed and freely moving rats provides further evidence that head fixation and voluntary motionlessness entail similar behavioral patterns.

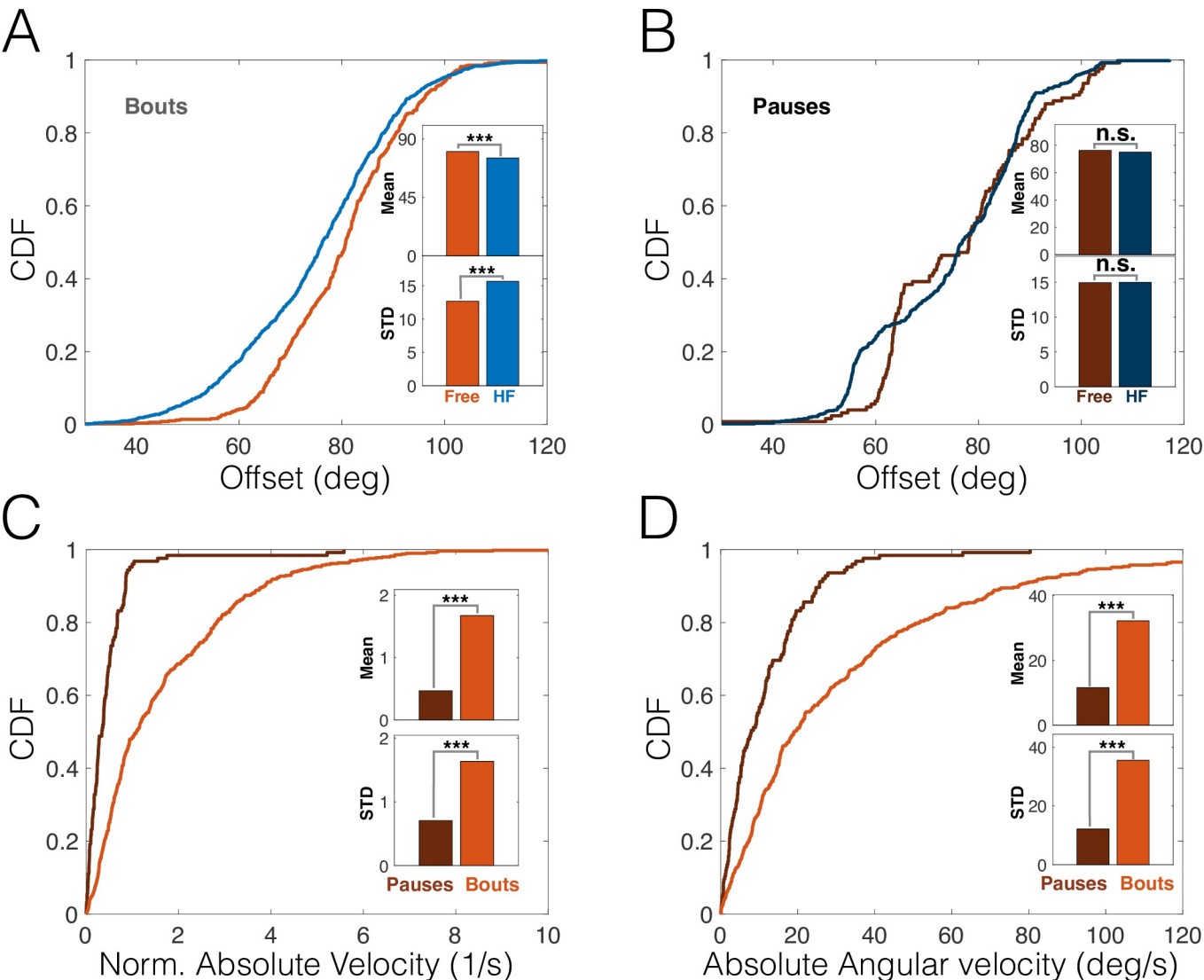

**Fig 8. (A–B) CDFs of whisking offset for HF (blue) and free (red) rats during bouts (A) and pauses (B).** (C–D) CDFs of absolute head velocity (C, normalized to head size) and angular velocity (D) for freely moving trimmed rats, during pauses (dark red) and bouts (light red). Insets: mean and STD of distributions. ***$p < 0.005$. The data and analysis code for this figure can be found here: https://github.com/avner-wallach/Rat-Behavior.git. CDF, Cumulative Distribution Function; HF, head-fixed; n.s., not significant; STD, Standard Deviation.

## Discussion

In this paper, we have shown that whisking kinematics predict consequent head and body locomotion and, consistently, that these kinematics depend on the behavioral context. The spatial and dynamical characteristics of rats' exploratory whisking were affected by the rats' ability to move and the number of whiskers they had available. It is commonly assumed that overt spatial attention is associated with preparing to move the body or the sensors towards a selected location or object [13,14,44–47] and, therefore, we suggest that the alterations in the spatial exploration described here reflect alterations in overt spatial attention. Our findings, therefore, suggest that both head-fixed rats and free but motionless ones dedicate more attention to whisker retraction than do rats in motion.

This interpretation suggests that the whisking pump, a subtle alteration in the whisking dynamics, is a useful indicator of perceptual attention. It was previously shown that, when induced by encountering an object, such pumps are robustly associated with object-oriented attention [13]. While free-air pumps were shown to have different temporal dynamics than those of TIPs [13], suggesting distinct underlying sensorimotor pathways, we demonstrated here that the 2 types of pumps share several key temporal and spatial features: temporal clustering, phase prolongation, spatial offset shifting, and bidirectional relationship with head motion. Critically, we show that free-air pumps predict the rats' future orienting behavior (compare Fig 1C in [13] and Fig 4E here). The differences in the temporal scales of sensorimotor kinematics [13] can probably be accounted for by differences in the underlying sensorimotor pathways. While TIPs seem to be implemented via brainstem loops [13,23,48], free-air pumps may involve higher-order sensorimotor loops.

The predictive relations between free-air pumps and the rat's locomotion may reflect the rat's expectation of future encounters. Thus, when the rat moves forward at moderate velocity while exploring the environment, it might expect novel encounters to occur mostly during protraction [26] and therefore increases the pump rate in that direction in order to improve sensory acquisition (Fig 3), either by extending protraction duration (Fig 2) or by the pump causing the whisker to briefly "revisit" places of interest. Conversely, when the rat is motionless, either by choice or because of imposed head-fixing, encounters may occur in either direction of whisker motion, and therefore, the rat shifts some of its spatial attention towards retraction (Figs 3 and 5). Importantly, this shift in spatial attention was also evident in the overall distribution of the whisking offset in head-fixed rats during active bouts, but not during pauses (Fig 8), which occur when the head is motionless and were hypothesized to reflect a passive receptive mode of attention [38].

These findings also offer an interesting way to reconcile an apparent contradiction in previously reported data. Our previous study in anesthetized animals on the sensory representation of whisker motion at the primary afferents and brainstem levels found cells responsive throughout the whisking cycle, with most cells responding to the protraction phase [49]. In awake head-fixed animals, however, brainstem, thalamic and cortical sensory cells show an overrepresentation of the retraction phase [50,51]. So far, no ethological or physiological explanation was given for this finding. A recent study [52] found correlations between the preferred phase of vibrissal afferents, and the activation of different facial muscle groups controlling whisking motion [53]. Whisker motion is evoked in anesthetized rats by stimulating the buccal motor branch of the facial nerve, which mostly innervates the intrinsic whisker pad muscles responsible for protraction [54,55] (the extrinsic nasolabialis muscle group involved in active retraction is innervated by the zygomatic branch). Thus, this method generates active protractions and passive retractions [56], and therefore, more cells responding to protraction were sampled. However, we can infer from the results reported here that awake head-fixed animals shift their attention backwards and may therefore strongly activate the muscle groups involved in controlled retraction [22], and the afferents correlated with that motion. It is also possible that increased attention to retraction involves activation of internal feedback loops within the brain [25,55], enhancing the activity related to this phase. In other words, the predominance of retraction related cells reported in awake head-fixed animals may reflect the behavioral context imposed by the experimental set-up, rather than the actual distribution of sensitivity in the vibrissal system. The conclusion arising from this possibility is far-reaching: behavioral context may bias physiological findings down to the cellular level, either via alteration of the sensorimotor interactions with the world at the periphery or via internal feedback loops in the central nervous system.

Our freely moving rats displayed a preferred whisking pattern, based on small amplitude whisks around a varying offset. This pattern appears to allow a combination of local active sensation with global (row-wide) passive reception, resembling in part the fovea–periphery division of work in vision. In contrast, trimmed rats applied large amplitude whisks around a fixed offset, probably in order to achieve similar spatial coverage with the few whiskers they had left. Overall, while freely moving rats used combinations of head and body movements to shift their attentional foci, head-fixed rats had to apply a wide range of whisking patterns, varying both amplitudes and offsets, possibly in an attempt to cover as many attentional foci as they could.

Animals use their available resources in an adaptive manner. We show here that rats adapt their perceptual behavior to their locomotion dynamics, both when voluntarily selected and when externally imposed, and suggest that they do so in a way that attempts to optimize coverage of the relevant space and future encounters with objects. Taken together, these observations are consistent with the closed-loop view of perception, in which essential perceptual variables, such as space coverage, are actively maintained when facing external or embodied constrains.

## Methods

### Whisking in freely moving rats

The experimental protocol is described in detail in [13]. Briefly, the whisking patterns of Wistar strain male albino rats aged 3–6 months were measured ($N = 3$ for free-air trimmed, $N = 4$ for free-air single row). On the day prior to behavioral recording, trimmed whiskers were clipped close to the skin (approximately 1 mm) under Dormitor anesthesia (0.05 ml/100 g, SC). All experimental protocols were approved by the Institutional Animal Care and Use Committee of the Weizmann Institute of Science.

Behavioral experiments were performed in a darkened, quiet room. The behavioral apparatus (see S1 Fig) consisted of a holding cage (25 cm width, 35 cm length, 29.5 cm height) with a small door (6.9 cm height, 6 cm width), through which the rats could emerge into the experimental area (18 cm × 20 cm) [57]. Both the holding cage and the experimental area were fixed approximately 15 cm above the surface of a table. The experimental area consisted of a back-lit Perspex plate with 1–2 objects (Perspex cubes and cylinders) placed on it. The experimental area was filmed from above by a high-speed, high-resolution camera (1,280 × 1,024 pix, 500 fps, CL60062; Optronis, Kehl, Germany). An in-house program (E. Segre, Weizmann Institute) triggered the high-speed camera whenever the rat emerged from the holding cage into the experimental area. Video recording stopped when the rat returned to the holding cage.

An experimental session consisted of recording a rat's whisking behavior whenever the rat was in the experimental area, over a period of 30–120 min. Preceding a session, the animal was placed in the holding cage for a 15-min acclimation period. During the acclimation period, the door of the holding cage was blocked. The experimental session began with unblocking the door to allow the animal to leave the holding cage and explore the experimental area at will. Each trial started when the rat moved from the holding cage to the experimental area and ended when the rat went back into the holding cage. The length of the experimental session varied depending on the animal's behavior, and the amount of recorded video. Whisker movements were tracked offline using the MATLAB (The MathWorks, Natick, MA, USA)-based WhiskerTracker image processing software (available at https://github.com/pmknutsen/whiskertracker/). Base angles of all existing macrovibrissa were tracked, as well as the location of both eyes and the tip of the nose. Only tracked segments that did not include object contacts were used for analysis in the current work.

## Whisking in head-fixed rats

The experimental protocol is described in details in [23]. Briefly, 7 male albino rats were head-fixed using screws glued to the skull under anesthesia. After full recovery, rats were gradually adapted to head fixation for 4–5 days. All except for 3 whiskers (C1, C2, and D1) on either side were clipped close (approximately 1 mm) to the skin during brief (5–10 min) isoflurane anesthesia and were retrimmed 2 to 3 times a week at least 2 hours before an experiment. A single experiment lasted up to 30 min but terminated earlier if rats showed signs of distress. Each rat had 1 or 2 experimental sessions a day, 2 or 3 times a week. All experiments were performed in a dark, sound-isolated chamber. Head orientation was estimated by imaging the corneal reflections of 2 infrared (880 nm) light-emitting diode (LED) spotlights. An imaginary line between the nose and eye on each side of the rat served as a reference line for the whisking angle. Bright-field imaging of the whiskers was accomplished by projecting infrared light (880 nm) with an array of 12 × 12 LEDs from below the animal. Video acquisition was triggered manually, and high-speed video was buffered and streamed to disks at either 500 or 1,000 frames/s. A total of 255 trials with an average duration of 8.3 ± 3.2 s (mean ± standard deviation, SD) were acquired. The 106 "free-air trials" and 149 "contact trials" were intermixed. Only the free-air trials are used for analysis in the current work. Whisker movements were tracked offline using the MATLAB (The MathWorks, Natick, MA, USA)-based *WhiskerTracker* image processing software (available at https://github.com/pmknutsen/whiskertracker/).

## Analysis

**Rhythmic decomposition and phase segmentation.** All analyses were performed on the base angle of whisker C2 on the left side of the snout. Analyses of head turns also used whisker C2 on the right side of the snout, and the results from both sides were pooled after mirroring. Signal was first filtered using a low-pass filter (sixth-order Butterworth, passband cutoff 20 Hz) to remove noise; this filtering was chosen to remove high-frequency noise due to tracking errors while preserving the pumping information. Whisking phase $\varphi$ was extracted using the Hilbert transform from a high-pass–filtered version (cutoff 4 Hz). Peaks and troughs were located within $\pm\pi/8$ of $\varphi = 0$ and $\pi$, respectively, and the signal was segmented into individual protraction and retraction phases. Offset of each motion was defined as the middle point between trough and peak; amplitude was half the distance between the 2 points.

**Bouts/pauses discrimination.** Histograms of the whisking amplitudes were computed using 200 logarithmic bins in the range 0.03–100 deg. These histograms were fitted using a GMM with 8 components. The maximum-likelihood threshold for discriminating between the 2 modes of the distributions was found by classifying the amplitude bins using the GMM and finding the first bin classified to one of the high-amplitude (mean > 3.16 deg) components.

**Pump detection and quantification.** Angular trajectory of each phase was differentiated to produce the angular velocity. Retraction velocity profile was negated. A pump was identified whenever the resulting velocity profile had more than one peak. The shapes of individual protraction/retraction phases were previously analyzed in great detail in [20]; there, 3 categories of motion were described: "single pumps," in which the protraction/retraction velocity profile contains a single peak (these are the "default," unmodulated whisking cycles); "delayed pumps," in which there are 2 velocity peaks, but there is no reversal in the direction of motion; and "double pumps," in which motion in the opposite direction (e.g., backward during protraction) is detected. We quantify the pump strength $\sigma_{pump}$ by using the formula $\sigma_{pump} = \frac{|v_{max} - v_{trough}|}{|v_{max}|}$, where $v_{max}$ is the maximal velocity during the protraction/retraction and $v_{trough}$ the velocity at the lowest trough. Note that by this definition, $\sigma_{pump} = 0$ for "single

pump" (i.e., no pump), $0<\sigma_{pump}<1$ for "delayed pump," and $\sigma_{pump}>1$ for "double pump." We did not observe any discontinuity in the distributions around $\sigma_{pump} = 1$, and therefore, delayed and double pump profiles were lumped together; we refer to these 2 profiles simply as pumps.

**Head motion.** Total head translational velocity in freely moving experiments was defined as the velocity of the middle point in between the 2 eyes (see S2A Fig). Head direction was defined as the direction of the line connecting this point and the tip of the nose. Length of this line was defined as head size used to normalize translational velocity (to units of heads/s). Direction of translational velocity was than subtracted from the head direction to obtain the translational direction in head-centered coordinates. The projection of this vector on the line pointing towards the nose was defined as thrust (longitudinal translational velocity), while the orthogonal projection was defined as slip (transverse translational velocity).

**Statistics.** Random permutations were used to evaluate significance of correlations (Figs 2A, 3, and 4) and run length distributions (Fig 2B); the bootstrap method was used in all comparisons between sets (Figs 5,6D and 7). Unless stated otherwise, 5,000 permutations/draws were used in each comparison. When comparing offset and amplitude distributions in different contexts (Figs 1B and 7), the samples of each set are not statistically independent because of temporal correlations [21,23]. Therefore, for each segment analyzed, we computed the autocorrelation functions of these variables and measured the lag (number of cycles) at which their significance dropped below the $p = 0.05$ level (using random permutations as control). The amplitude/offset sequence was then diluted by this lag to obtain statistically independent samples.

**Whisking envelope entropy.** To estimate the dispersion of the whisking envelope (Fig 6D), we measured the bivariate probability distribution $p(\theta_{amp},\theta_{off})$ by binning each variable into 25 bins. We then measured the information entropy using the formula $H = \sum_{i=1}^{25} \sum_{j=1}^{25} -log(p(\theta_{amp} = x_i, \theta_{off} = y_j))p(\theta_{amp} = x_i, \theta_{off} = y_j)$. To enable statistical comparison between contexts, random sets of identical size from all data sets were required. Seventy-five different collections of tracked segments were randomly generated from each data set so that the total duration of each collection was approximately 30% of the total length of the smallest data set. Entropy was then calculated for each of the resulting subsets to produce a distribution of entropies for each data set (box plots, Fig 6D). Entropy was also calculated for each of the data sets in its entirety (inset in Fig 6D).

## Supporting information

**S1 Fig. Freely moving apparatus.** The apparatus included a holding cage with door (left) approximately 15 cm above the surface of a table, the experimental area (back-lit Perspex plate with acrylic cubes and cylinders), and a high-speed, high-resolution overhead camera. (TIF)

**S2 Fig. Head-motion analysis.** (A) Head tracking scheme. The 2 eyes and the tip of the nose were tracked in each video frame. Midpoint between eyes was defined as the head center. The azimuth of the line connecting this point and the nose is the head direction $\alpha$, while the length of this line is the head size $d$. The time derivative of the head direction, $d\alpha/dt$, is defined as turn (head rotation). The time derivative of the head location is the head velocity $V$, which has a longitudinal component thrust ($V_{th}$) and a transverse component termed slip ($V_{sl}$). These components were normalized to the head size $d$ and so are presented in units of heads/s. (B) Joint probability density of the thrust and turn variables. Pearson coefficient R = −0.035. Dashed white line: linear regression. (C) Joint probability density of the slip and turn variables. Pearson coefficient R = 0.863. Dashed white line: linear regression. The data and analysis code

for this figure can be found here: https://github.com/avner-wallach/Rat-Behavior.git.
(TIF)

**S3 Fig. Robustness of pump-rate indication prior to change of motion.** Abscissa in all panels: threshold of used to identify individual motions (turn/thrust). Gray shading in all panels: distributions of random-permutation–generated controls. (A–B) The protraction pump rate (normalized to median control levels) 100 ms prior to change of motion, computed for various levels of detection threshold. (A) Turn. Solid line with filled circles: towards pumping side; dashed line with asterisks: away from pumping side. Elevation in pump probability prior to turning towards pump is significant throughout the threshold range. (B) Thrust. Solid line with filled circles: onset of motion forward; dashed line with asterisks: onset of motion backwards. Elevation in probability of forward onset is significant throughout the threshold range. (C–D) The turn probability (normalized to median control levels) 100 ms following a pump, computed for various levels of detection threshold. Solid line with triangles: towards pumping side; dashed line with upside-down triangles: away from pumping side. (C) Protraction pump. Elevation in probability of turning towards pump is significant throughout the threshold range. (D) Retraction pump. The data and analysis code for this figure can be found here: https://github.com/avner-wallach/Rat-Behavior.git.
(TIF)

## Acknowledgments

We thank Sahar Froim for experimental assistance and Mitra Hartmann and Mathew Diamond for discussions and advice.

E.A. holds the Helen Diller Family Professorial Chair of Neurobiology.

## Author Contributions

**Data curation:** David Deutsch, Tess Baker Oram.

**Formal analysis:** Avner Wallach.

**Funding acquisition:** David Deutsch, Tess Baker Oram, Ehud Ahissar.

**Investigation:** Avner Wallach, David Deutsch, Tess Baker Oram, Ehud Ahissar.

**Methodology:** Avner Wallach, David Deutsch, Tess Baker Oram.

**Resources:** Ehud Ahissar.

**Software:** Avner Wallach.

**Supervision:** Ehud Ahissar.

**Visualization:** Avner Wallach.

**Writing – original draft:** Avner Wallach.

**Writing – review & editing:** Avner Wallach, David Deutsch, Tess Baker Oram, Ehud Ahissar.

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
