## [Editor Report · Decision Letter 0]

30 Oct 2019

Dear Dr Wallach, 

Thank you for submitting your manuscript entitled "Predictive whisker kinematics reveal context-dependent sensorimotor strategies" for consideration as a Research Article by PLOS Biology.

Your manuscript has now been evaluated by the PLOS Biology editorial staff, as well as by an academic editor with relevant expertise, and I'm writing to let you know that we would like to send your submission out for external peer review.

Please re-submit your manuscript within two working days, i.e. by Nov 01 2019 11:59PM.

Kind regards,

Roli Roberts

Senior Editor

PLOS Biology

---

## [Decision Letter · Decision Letter 1]

13 Dec 2019

Dear Dr Wallach,

Thank you very much for submitting your manuscript "Predictive whisker kinematics reveal context-dependent sensorimotor strategies" for consideration as a Research Article at PLOS Biology. Your manuscript has been evaluated by the PLOS Biology editors, an Academic Editor with relevant expertise, and by three independent reviewers. I should point out that the Academic Editor who was involved in our original decision to review was not able to continue helping us with your paper, so we sought advice from a different Academic Editor for the decision.

You'll see that two of the reviewers find your claims to be of interest, and two are of the opinion that these claims are largely well-supported. However, reviewer #2 raises a number of significant concerns about your analyses and the strength of support for several of your central claims, and these will have to be thoroughly addressed for us to consider your study further.

In light of the reviews (below), we will not be able to accept the current version of the manuscript, but we would welcome re-submission of a much-revised version that takes into account the reviewers' comments. We cannot make any decision about publication until we have seen the revised manuscript and your response to the reviewers' comments. Your revised manuscript is also likely to be sent for further evaluation by the reviewers.

We expect to receive your revised manuscript within 2 months. 

**IMPORTANT - SUBMITTING YOUR REVISION**

*NOTE: In your point by point response to to the reviewers, please provide the full context of each review. Do not selectively quote paragraphs or sentences to reply to. The entire set of reviewer comments should be present in full and each specific point should be responded to individually, point by point.

*Re-submission Checklist*

*Published Peer Review*

*PLOS Data Policy*

*Blot and Gel Data Policy*

Sincerely,

Roli Roberts

Senior Editor

PLOS Biology

REVIEWERS' COMMENTS:

Reviewer #1:

In this manuscript, Wallach and colleagues study how whisking kinematics depend on the rat’s behavioural context. They characterise whisking at high temporal resolution in two experimental contexts – (1) when the rat was able to move its head versus being head-fixed and (2) when different number of whiskers were available. In freely moving rats, whisking kinematics predicted consequent head and body motion; in particular, the free-air pump, a subtle alteration in the whisking dynamics, predicted the rats’ future orienting behaviour. The authors also demonstrate that whisking kinematics change systematically with the number of whiskers available and with head fixation. The results further support the idea that whisking pump is an indicator of perceptual attention. One interesting conclusion from the findings is that head-fixed rats (and free but motionless ones) dedicate more attention to whisker retraction than do rats in motion. 

This is a nice paper, it is timely, and addresses a fundamental biological question. The paper reports precise quantification of a sensorimotor action under experimental control, with analyses that are elegantly performed. I find the results interesting and the paper suitable for publication. I have two comments to improve the presentation and discussion of the data.

1. At the core of the paper is the analysis related to the whisker pump. However, the concept is only briefly introduced without giving it enough context. The pump is defined as:

“It was noted in several previous studies that the velocity profile of individual phases is occasionally multi-peaked, a feature termed ‘pump’.” It would be useful to give a more quantitative definition of the pump in the results section, and elaborate why these sensorimotor twitches are thought to relate to attention. 

2. It would be helpful to make a distinction between attention that requires exploratory whisking behaviour (e.g. when rats extend their whiskers to palpate an object) and attention that requires whiskers to remain stationary. For example, in detecting vibrations that are generated from an approaching predator, rats might immobilise their whiskers in the receptive mode (see Diamond and Arabzadeh, 2013) to enhance signal to noise for detection of minute vibrations that originate from the environment (rather than from whisker object interaction). In such settings, there might be periods of high spatial attention that require suppression of exploratory whisker movement. The authors could speculate on what features of whisking kinematic profile help distinguish between these forms of attention.

Reviewer #2:

Wallach et al. investigated the contingency of whisking kinematics on the animal’s behavioral context that arises from both internal processes and external constraints. The authors identify an interesting, oft noted, but under-explored field that deals with how vibrissal behavior is affected by behavioral context. Namely, they compared different aspects of vibrissal behavior measured in three contexts: 1. Head-fixed rats with a trimmed whisker pad; 2. freely moving rats with three whisker configuration; 3. freely moving rats with an entire single row of whiskers. They focus their analysis on a kinematic feature called the free-air pump. The authors, use high speed filming of head and whiskers in which they analyze behavioral consequences of different conditions. 

The overall conclusion of the work is that rats adapt their active exploratory behavior in a "homeostatic" attempt to preserve sensorimotor coverage under changing environmental conditions and changing sensory capacities, including those imposed by various laboratory conditions. The authors base this conclusion on the following main points of evidence that: 1) pump behavior during both voluntary motionlessness and imposed head-fixation exposed a backward redistribution of sensorimotor exploratory resources; 2) head-fixed rats employed a wide range of whisking profiles to compensate for the loss of head and body-motor degrees of freedom; 3) changing the number of intact vibrissae available to a rat resulted in an alteration of whisking strategy consistent with the rat actively reallocating its remaining resources. 

However, the strength of the evidence shown in this work is not commensurate with the level of the claims being made, and the different experiments do not complement each other in order to draw a unified conclusion. Moreover, there are significant methodological concerns that draw into question the validity of the separate conclusions, and there are choices in the presentation of the data that obfuscate the conclusions being drawn.

Major Comments:

1. Ln. 88. "All analyses were performed on the protraction angle of whisker C2" Why only C2 and why protraction only. 

2. Lns. 95-96. 1B); "the 2 deg threshold was set based on the 96 bimodal distribution of whisking amplitudes (see Fig S2)". What is the statistical criterion for setting the threshold?

3. Lns. 97-98. " while freely moving rats 98 whisked 89.8% of the tracked time (inset, Fig 1B)". How is the text related to the insert?

4. Define CFD in the text.

5. Lns. 105-106. " All analyses described below were performed on bout epochs (i.e., non-whisking epochs were excluded). what's the point in showing one w/o the other. Especially when they are related. Should at least compare them.

6. Lns. 107-108. stages: "protraction, 107 in which the whiskers move rostrally, and retraction, in which they move caudally". What were the criterions to distinguish between these two epochs? 

7. Ln. 113. "Next we compare various properties of ‘free-air pumps’ and TIPs". What were the criterions to define fap and tips?

8. Lns. 129-130. " TIPs are longer and shifted forward when compared with protractions with no pumps." Should show this by putting some reference line.

9. Lns. 137-139. Conclusions that has to be shown here. Elaborate how you got to this conclusion from you data. 

10. Ln. 147. Why the reference whisker. If everything is correlated why look at a specific whisker? Is the analysis done on all whiskers?

11. Fig. 3. Significant values compared to what? Not clear.

12. Lns. 154-156. Conclusion. Elaborate how you got to this conclusion from you data. 

13. Ln 158. This is another example for a claim that has no relevance to the data. They assert that head movement is a change in the animal’s exploratory behavior. 

14. Lns. 176-178. Conclusion. Yet another far-reaching conclusion.

15. Fig. 6. The conclusion is that head-fixed animals with missing whiskers are whisking differently from freely roaming animals. This is the take home message.

16. Lns. 282-284. " The facial nerve stimulation method.." The claim is not true. Both types of muscles are activated during this paradigm.

Reviewer #3:

This manuscript presents an analysis of the whisking behavior of rats in relation to their head motion under different experimental contexts (i.e., freely moving vs. head-fixed and full-row of whiskers available vs. just three whiskers available). The main experimental finding are that some specific whisking patterns (named pumps) are correlated with (and thus predictive of) rat head direction during exploratory movements. In addition, the distribution of these pumps during protractions and retraction phases (i.e., forward and backward movements of the whiskers) is very different in head-fixed animals, as compared to freely exploring rats, but it becomes similar during voluntary motionlessness in free animals. Finally, head-fixed rats display wider-range whisking movements, whose amplitude becomes similar to that of freely moving rats only if head displacements are also taken into account, thus suggesting that the animals actively compensate for the reduced mobility of the head by resorting to a wider range of whisking patterns.

Overall, the manuscript is well organized and well written, the study is solid and interesting, given that the a better, quantitative knowledge of whisking behavior in relation to locomotion can be important for future neurophysiological studies of somatosensory representations. I wonder, however, how widespread and general can be the interest for such study by the readership of Plos Biology. While it seems to me that this is a well-crafted study, surely suitable for a more specialized journal, I am not sure it meets the requirements of generality and far-reaching impact of this journal.

Beside this general consideration, I have a few specific comments/criticisms.

1) Fig. 1B. Not clear the color code here. Which of the traces is the actual (measured) protraction angle? The legend says the gray line, but the gray line seems a perfect sinusoid (so it looks like it is the carrier wave, the fast oscillation, rather than the measured signal). It looks like the protraction angle is the black trace, but the legend says it is the offset. Please clarify.

2) Fig. S2. It is not a good idea to show these largely overlapping distributions as bar plots. Better to use line plots, so that the profile of all 4 distributions become visible.

3) Lines 104-105. Why for free moving rats it was hard to collect complete bouts? This should be explained.

4) Fig. 4. It is unclear to me why in panels A-D of this figure the gray curve indicating the thrust velocity or angel velocity is not at zero when the time of change is zero. Isn't the time of change defined as the time at which the motion (either forward thrust or angular) reverts its direction? Shouldn't then be zero the velocity at this time?

-----------------

---

## [Editor Report · Decision Letter 2]

6 Apr 2020

Dear Dr Wallach,

Thank you for submitting your revised Research Article entitled "Predictive whisker kinematics reveal context-dependent sensorimotor strategies" for publication in PLOS Biology. I have now obtained advice from the Academic Editor, who has thoroughly assessed your responses to the reviewers' comments. Please accept my apologies for the delay incurred during these challenging times.

We're delighted to let you know that we're now editorially satisfied with your manuscript. However before we can formally accept your paper and consider it "in press", we also need to ensure that your article conforms to our guidelines. A member of our team will be in touch shortly with a set of requests. As we can't proceed until these requirements are met, your swift response will help prevent delays to publication.

IMPORTANT: Please also make sure to address the Data Policy and other policy-related requests noted at the end of this email.

*Copyediting*

*Published Peer Review History*

*Early Version*

*Submitting Your Revision*

Sincerely,

Roli Roberts

Senior Editor

PLOS Biology

ETHICS STATEMENT:

-- Please include the full name of the IACUC/ethics committee that reviewed and approved the animal care and use protocol/permit/project license. Please also include an approval number.

-- Please include the specific national or international regulations/guidelines to which your animal care and use protocol adhered. Please note that institutional or accreditation organization guidelines (such as AAALAC) do not meet this requirement.

-- Please include information about the form of consent (written/oral) given for research involving human participants. All research involving human participants must have been approved by the authors' Institutional Review Board (IRB) or an equivalent committee, and all clinical investigation must have been conducted according to the principles expressed in the Declaration of Helsinki.

DATA POLICY:

Regardless of the method selected, please ensure that you provide the individual numerical values that underlie the summary data displayed in the following figure panels as they are essential for readers to assess your analysis and to reproduce it: Figs 1, 2, 3, 4, 5, 6, 7, 8, S2, S3. NOTE: the numerical data provided should include all replicates AND the way in which the plotted mean and errors were derived (it should not present only the mean/average values). I note your intention to deposit the data in Columbia University’s Academic Commons repository; please do so and supply the URL so that we can assess your compliance.

---

## [Editor Report · Decision Letter 3]

11 May 2020

Dear Dr Wallach,

On behalf of my colleagues and the Academic Editor, Rony Azouz, I am pleased to inform you that we will be delighted to publish your Research Article in PLOS Biology. 

Early Version

PRESS 

Kind regards,

Alice Musson

Publishing Editor, 

PLOS Biology

on behalf of

Roland Roberts,

Senior Editor

PLOS Biology